# HRV Monitoring Using Commercial Wearable Devices as a Health Indicator for Older Persons during the Pandemic

**DOI:** 10.3390/s22052001

**Published:** 2022-03-04

**Authors:** Eujessika Rodrigues, Daniella Lima, Paulo Barbosa, Karoline Gonzaga, Ricardo Oliveira Guerra, Marcela Pimentel, Humberto Barbosa, Álvaro Maciel

**Affiliations:** 1Department of Physiotherapy, Federal University of Rio Grande do Norte—UFRN, Natal 59072-970, Brazil; ricardo.guerra@ufrn.br (R.O.G.); marcela.pimentel.094@ufrn.edu.br (M.P.); alvarohuab@ufrnet.br (Á.M.); 2Department of Computer, Center for Strategic Health Technologies—NUTES, Campina Grande 58429-500, Brazil; daniella.lima@nutes.uepb.edu.br (D.L.); paulo.barbosa@nutes.uepb.edu.br (P.B.); karolineandrade179@gmail.com (K.G.); 3Laboratório de Análise e Processamento de Imagens de Satélites (LAPIS), Instituto de Ciências Atmosféricas, A. C. Simões Campus, Universidade Federal de Alagoas, Maceió 57072-900, Brazil; barbosa33@gmail.com

**Keywords:** heart rate variability, remote monitoring, wearables

## Abstract

Remote monitoring platforms based on advanced health sensors have the potential to become important tools during the COVID-19 pandemic, supporting the reduction in risks for affected populations such as the elderly. Current commercially available wearable devices still have limitations to deal with heart rate variability (HRV), an important health indicator of human aging. This study analyzes the role of a remote monitoring system designed to support health services to older people during the complete course of the COVID-19 pandemic in Brazil, since its beginning in Brazil in March 2020 until November 2021, based on HRV. Using different levels of analysis and data, we validated HRV parameters by comparing them with reference sensors and tools in HRV measurements. We compared the results obtained for the cardiac modulation data in time domain using samples of 10 elderly people’s HRV data from Fitbit Inspire HR with the results provided by Kubios for the same population using a cardiac belt, with the data divided into train and test, where 75% of the data were used for training the models, with the remaining 25% as a test set for evaluating the final performance of the models. The results show that there is very little difference between the results obtained by the remote monitoring system compared with Kubios, indicating that the data obtained from these devices might provide accurate results in evaluating HRV in comparison with gold standard devices. We conclude that the application of the methods and techniques used and reported in this study are useful for the creation and validation of HRV indicators in time series obtained by means of wearable devices based on photoplethysmography sensors; therefore, they can be incorporated into remote monitoring processes as seen during the pandemic.

## 1. Introduction

Heart rate variability (HRV) consists of the periodic oscillation between heartbeats, and these oscillations are modulated by the Autonomic Nervous System [1,2]. Over the years, several studies have been carried out seeking to correlate the individual’s age group with the autonomic changes in the heart, bringing evidence that human aging can promote a decrease in vagal tone and, consequently, in heart rate variability [2,3].

One of the results of the aging process is the decline in the ability of the autonomic nervous system to adjust blood flow in response to external situations. This autonomic control can be assessed and monitored in a non-invasive way through the heart rate variability being considered as a parameter to estimate the interaction between the brain and the cardiovascular system in elderly individuals [2].

Changes in HRV patterns provide a sensitive and early indicator of possible declines in health. Efficient autonomic mechanisms provide a high HRV, which is a sign of good adaptation to intrinsic and extrinsic factors, characterizing a healthy individual. Conversely, low HRV reflects an abnormal and insufficient adaptation of the autonomic nervous system, which may indicate the presence of physiological changes relevant to the individual’s health. It is necessary to monitor and investigate the reasons why this variability is altered [4].

Therefore, HRV can be considered a promising early biomarker of cognitive impairment in the older population. This index must be evaluated within a preventive perspective to minimize the impacts caused by aging and the risks of developing cognitive impairment [2,5].

Traditionally, HRV is measured in clinical or laboratory settings, using equipment such as the electrocardiogram (ECG), cardiac belt, or Holter. These tools have some characteristics that prevent their use in day-to-day life and hinder the continuous and routine monitoring of HRV. In general, problems of biocompatibility, discomfort, unsatisfactory design, and limited access to clinical or laboratory environments are factors that make the use of these instruments less appropriate for health assessments and unlikely to be adopted in the individual’s daily routine [6,7,8].

On the other hand, sensors for capturing physiological data in wearable devices emerged quickly in the market and became popular among its users and in the field of scientific research, due to technological advances of companies that stand out in this market of wearable devices, focusing mainly on the creation of smartwatches [9].

The combination of wearable devices and health systems has proven over the years to be a viable strategy for resolving health conditions efficiently, characterizing a new era for the monitoring, diagnosis, treatment, and prevention of many diseases, through the detection of early patterns of potential health declines [9,10].

The appearance of wearable devices with photoplethysmography sensors make it possible to capture the cardiac pulse in minutes. However, due to manufacturers’ design decisions, wearable device APIs provide time series of heart rate with missing data, compromising the achievement of heart rate variability indicators, considering that the acquisition of HRV information is performed through identification of heartbeats every second [11,12].

The goal of this study is to introduce an intelligent model for HRV based on time series provided by commercial wearable devices. In this study, the chosen devices were from the Fitbit brand. This was achieved through the introduction of the Sênior Mobile Health (SMH) platform (seniorsaudemovel.com.br accessed on 23 January 2022), which has features to monitor the health and wellbeing of older persons and is being experimented in several pilot studies in Brazil. Among these features, we also explored smart algorithms to build a model that aims to provide accurate HRV parameters trying to overcome the current technological limitations of a large scale commercially available wearable device. We defined a workflow that involves data pre-processing, modeling, and model’s execution.

The next sections of this study are structured as follows: Section 2 introduces the materials and methods of the work, with the description of the SMH platform and its algorithms, components, and architectural decisions. We detail the approach to investigate the feasibility of HRV as a health indicator using wearable. Section 3 details the validation results at different layers of investigation of the HRV model and different elderly data. Section 4 discusses the main achievements of this work. Finally, Section 5 presents the conclusions and future works.

## 2. Materials and Methods

In this section, we discuss the method of developing the remote monitoring system and the strategy used to assess the health conditions of the older person using HRV as the main health indicator. First, we present the chosen device and the advantages found for the acquisition of the sample’s physiological data. In the second topic of this section, we discuss the HRV metrics that were chosen for this study. In the third topic, we approach the developed remoted monitoring system, i.e., the SMH platform. In the fourth topic of this section, we describe the eligibility criteria for recruiting the sample and procedures adopted for data acquisition. Finally, in the fifth topic, we seek to establish the data analysis workflow, which supports the next section on the results of the assessment of the HRV algorithm’s effectiveness and health conditions of the monitored older people during the pandemic, taking into account the variables of sleep quality, daily steps, and active minutes, considered in the present study as independent variables and HRV as the dependent variable.

The overall method is summarized as follows: Regarding the quality of data pre-processing, we ensured that missing data were filled in the best way. In this sense, we tested several different conventional time series imputation methods comparing the time series of Fitbit smart watch (Fitbit Company, San Francisco, CA, USA) and Polar H10 (Polar Electro Brasil Comércio, Distribuição, Importação e Exportação Ltd.) heart rate monitor chest strap. After identifying the best filling approach, the Piecewise Cubic Hermite Interpolating Polynomial method (PCHIP), the modeling approximated the data obtained from Fitbit to the data coming from Polar H10. Therefore, we compared different prediction methods and the best one was the Kamath Rule, with similar results to the chosen gold standard, and a performance better than other methods. Finally, in the model’s execution, our *t* statistics show satisfactory results for the standard deviation of the average of all normal RR intervals, expressed in milliseconds (SDNN), square root of the square mean of the differences between adjacent normal RR intervals, expressed in milliseconds, that is, the standard deviation of the differences between adjacent normal RR intervals (RMSSD), and the percentage of adjacent RR intervals with a difference in duration greater than 50 milliseconds (pNN50), which are common HRV scores; being identical or smaller than the critical value, the *p*-value was greater than the significance level meaning that there is no change in HRV in HRV parameters when using the SMH. Since this study was motivated by the COVID-19 pandemic scenario that imposed isolation constraints to elderlies, these results show that commercial wearable devices combined with the SMH platform can be used as an effective tool for HRV monitoring.

### 2.1. Wearable Device

For the present research, the Fitbit brand device, model Inspire HR, was used. This device is already widely available on the market and meets important criteria such as biocompatibility of the material; thus, reducing the possibility of allergic processes due to constant contact with the skin. Furthermore, criteria such as comfort, design and usability are provided by these devices, which are considered important factors in the adhesion to the solution.

Another important aspect found on Fitbit is that there are incentives for researchers to use their devices in clinical studies worldwide, as the company makes available the data of users who have provided consent to researchers, both in the form of summary data and time series. The mechanisms of automatic wireless synchronization with other devices are also very practical for the daily use of the elderly and their caregivers. Finally, an open Application Programming Interface (API) is available, providing the possibility to create new code and functionalities for third-party applications or even new applications for some versions of their products.

The provision of open APIs is a great advantage for this research, allowing the integration of data coming from the different sensors to the developed platform, granting a monitoring aligned to the desired goals. Not all devices on the market make their APIs open and available, making it impossible to obtain data from sensors and analysis. This was one of the main criteria for excluding other wearables.

The Fitbit Inspire HR data are transmitted via Bluetooth Low Energy (BLE) to the wearable device provider smartphone application (Fitbit app), which makes the necessary data processing and sends it to the wearable device provider server (Fitbit server), where it becomes available for consultation by third parties via web services. Due to the manufacturer’s internal policies and security measures, third-party applications are not allowed to collect data directly from the wearable device; therefore, the sole party responsible for the collection is the wearable device provider application.

After data are collected by the manufacturer’s application and made available on its server, third-party applications can collect these data, if they have the appropriate permissions. We used these data in a mobile application and web dashboard, providing smart algorithms to predict adverse health conditions and promote real-time and continuous monitoring of elderly health. Such algorithms are currently analyzing, besides HRV (which is the main focus of this study), steps, nocturia, quality of sleep, and gait speed, which are closely linked to adverse effects in the health of older people.

### 2.2. HRV Metrics

Most biological signals are defined as quasiperiodic, which means that they can vary repetitively at almost regular time intervals. In this sense, a periodic variable can be analyzed as a function of time or as a function of the frequency at which the event takes place. Therefore, an HRV analysis can be performed in the time domain and in the frequency domain. In this study, we implemented algorithms for HRV metrics only in the time domain due to lesser complexity and pragmatism for delivering the SMH platform [13,14].

For the HRV analysis in the time domain, so called because it expresses the results in time units (milliseconds), each normal RR interval (heartbeat) is measured during a certain time interval and, from there, based on statistical or geometric methods, we calculate the metrics that reflect fluctuations in the duration of cardiac cycles [4,13].

The statistical metrics used to assess HRV can be divided into two categories: metrics based on the measurement of RR intervals individually (SDNN, SDANN, and SDNNi) and metrics based on the comparison between two adjacent RR intervals (pNN50 and RMSSD) [13].

The statistical metrics pNN50 and RMSSD predominantly represent vagal tone [1,4,13,15].

The metrics based on the measurement of RR intervals individually, such as SDNN, SDANN (standard deviation of the means of normal RR intervals every 5 min, expressed in milliseconds), and SDNN index (average of the standard deviations of the normal RR intervals every 5 min, expressed in milliseconds) depict the global variability, comprising the parasympathetic and sympathetic autonomic nervous system [1,4,13,15].

### 2.3. The SMH Platform

Through the SMH platform it is possible to carry out continuous, remote, and real-time monitoring of the data acquired from the Fitbit Inspire HR of the sample participants, monitoring the data of interest in the present study.

The development of the SMH platform was established with services related to user and information management, data science, and IoT. Artificial intelligence algorithms and time series analysis were employed to monitor important data on the health of the older person.

Fitbit brand devices offer data through time series of active minutes, number of daily steps, heart rate measurements and sleep stages. These data are obtained through the APIs and integrated into the SMH platform, where they are stored, analyzed, and displayed by means of graphs, indexes, and values. The next figures show features in the mobile app and in the dashboard related to the HRV monitoring. All the figures show fictitious data because the purpose is just to illustrate the functionalities.

Figure 1 shows the role of mobile app when monitoring HR in two different situations. Figure 1a is a walk screen and Figure 1b is a sleep screen. Indicators with measurements of positions and dispersion are available to visually correlate to other system features such as zones classification, sleep awakenings and much more. The displayed HR signal and its features are the main input for the HRV smart algorithms.

Figure 2 shows the SMH web dashboard with a general overview of functionalities. Besides the HRV indicator in the upper middle part of the figure, which currently displays the SDNN, we also have indicators of activities, walking speed, steps, sleep quality, night awakenings, distance, active minutes, calories, heart rate and other external measurements besides the Fitbit Inspire sensors measurements, such as weight, BMI, among others.

Now we focus more specifically on the HRV feature in the SMH dashboard. Figure 3 shows the history of heart rate monitoring, displaying centrality metrics, such as the minimum, maximum and average HR, as well as time domain metrics such as SDNN, SDNN index, and RMSSD.

Figure 4 presents an excerpt of the SMH platform REST API, using the Swagger tool (swagger.io, (accessed on 23 January 2022)), with three important routes that compose the HRV service. The first route, GET/metrics, retrieves all the existing HRV metrics in the database, regardless of the patient. The second route, GET/patients/{id}/metrics, retrieves all the HRV metrics for a specific patient. In this route, we can also filter the HRV metrics by period. Finally, the third route, DELETE/patients/{id}/metrics/{date}, allows to delete the HRV metrics provided for a specific day.

Figure 5 details the metrics data model that contains all the HR and HRV features collected. This was defined and documented using the Swagger tool for documentation of REST APIs. Each line contains the name of the metrics, its type (e.g., string or number) and description.

### 2.4. Sample Description and Solution Application

For the acquisition of data from the heart rate time series and construction of intelligent algorithms, 10 older persons were recruited.

The study was conducted with community-dwelling elderly people aged 60 years or over, residents on the city of Campina Grande (Brazil). The sample was recruited for convenience and includes active individuals without chronic-degenerative diseases and individuals with neurofunctional disorders such as Parkinson’s and Alzheimer’s or sequelae caused by stroke. Individuals with cognitive impairment were assessed by the Leganés Cognitive Test following scores below the cutoff level for dementia (≤22 points), those who did not have the ability to use a smartphone, those with severe visual problems, and individuals bedridden were excluded from the sample. Data collection was carried out from May 2020 to November 2021. All participants were informed of the research objectives and procedures and signed the Free and Informed Consent Form. The study protocol was approved by the Ethics and Research Committee of the State University of Paraiba (approval number: 34702520.2.0000.5187).

The data collection procedure was performed with the elderly positioned in a chair with an arm and backrest for the spine, in an environment with controlled temperature and without any visual or sound stimulus. Participants were asked to remain in a sitting position for about 15 min.

The heart rate data acquisition was obtained through the Polar H10, which is recognized as the gold standard instrument for classification of HRV. At the same time, the research participants used the Fitbit Inspire HR in the left wrist region, located two fingers above the styloid process of the ulnar bone. Both devices were placed in constant contact with the skin during the signal acquisition process.

All collected data were sent to the SMH platform for storage, and then the data analysis and processing procedures, construction of intelligent models and deployment process were carried out.

### 2.5. Development Process Workflow

The process of building the intelligent model of the SMH platform is presented in Figure 6 and Figure 7. Figure 6 shows an overview of the smart model and Figure 7 exhibits the detailed workflow for the development process of the three main steps defined in Figure 6: (i) data pre-processing, which mainly details the process of cleaning and dealing with missing data; (ii) modeling, which focus on the process of developing, training and evaluation of the smart model; and (iii) model’s execution, that describes the deployment process.

As shown in Figure 7, the SMH platform receives HR data from the Fitbit Inspire HR and transforms it into HRV information. Commercial wrist-worn wearables, such as Fitbit, measure HR through a photoplethysmography (PPG) sensor which uses a pulse oximeter to illuminate the skin and calculate changes in light absorption, detecting oscillations in blood volume during heart activity. The reference standard signal used for monitoring HR is provided by electrocardiogram (ECG) sensors, which directly measure the electrical signals produced by heart activity, being able to capture more comprehensive signals than the PPG sensors present on wearable devices. However, under ideal circumstances, considering healthy subjects at rest, PPG provides accurate Inter-Pulse Intervals (IPI), indicating that these devices may be used as an alternative to ECG for HRV analysis [16].

According to [16,17] HR monitors such as the ones produced by Polar provide heart rate measurements as precise as the ones of clinical ECG monitors, considering subjects at rest. Furthermore, when considering long-term monitoring of individuals freely moving, [18] demonstrated that Polar H10 is more accurate than a Holter monitor, the standard portable device for measuring HR [19].

To mitigate the inaccuracy of PPG sensors, which may be caused by a number of different factors (e.g., skin tone, obesity, age, gender, respiration, venous pulsation, body temperature, motion artifacts, and ambient light) [20], we adopted a method that seeks to approximate the data obtained from wearable devices to the data coming from ECG sensors, providing a dataset composed of IPI as inputs and the Inter-Beat Interval (IBI) as outputs, assuming that there is an unknown underlying function that consistently maps inputs to outputs.

#### 2.5.1. Data Pre-Processing

Before trying to learn the mapping function, we needed to assess other issues inherent from the data. The Instantaneous Heart Rate (IHR) data acquired through Fitbit consists of a time series that presents gaps (missing data) due to possible measurement failures. In order to guarantee the integrity of the data, it is necessary to fill in those gaps. The analysis of certain properties of the time series, such as its stationarity, provides an indication of possibly appropriate methods for the filling process. According to the Augmented Dickey Fuller Test (ADF Test), a common statistical test used to verify whether a provided time series is stationary or not, the series is non-stationary; implying that mean, variance, and autocorrelation are not constant over time, as depicted in Figure 8.

For that reason, methods such as replacing the missing values with the mean of the series would probably provide a poor result, as can be checked in Figure 9.

To ensure data were filled in the best way, we tested several different conventional time series imputation methods. In order to measure the performance of each approach, missing values were randomly introduced to a complete time series, obtained from Polar H10 and filled in with the different methods comparing the predicted results with the original values:Mean filling: mean filling replaces the gaps with the mean value of the series;Forward fill: this method fills in the missing values by propagating the last valid observation forward;Backward fill: this approach fills the missing values of the data frame backwards;Linear interpolation: the linear interpolation method fits a line of polynomials to the end points of the gap using a linear formula to estimate the missing values;Quadratic and cubic spline interpolation: Type of interpolation in which the interpolant is a special type of piecewise polynomial named spline. This method fits lower degree polynomials, in this case quadratic or cubic, to a small subset of values rather than fitting only one polynomial with high degree to all the values;Piecewise cubic Hermite interpolating polynomial (PCHIP): PCHIP consists of a spline third-degree piecewise polynomial function specified by its values and first derivatives at the end points of the corresponding domain interval;Akima interpolation: The Akima interpolation is a continuously differentiable spline interpolation, formed from piecewise third order polynomials, in which data from the next neighbor points is used to determine the coefficients of the polynomial;Exponentially weighted moving average (EWMA): EWMA estimates the missing values as a weighted average of the historical observations, giving older observations lower weights that decrease exponentially as the data points age;Interpolation by nearest neighbor: in this method, the imputed value is equal to that of the closest known point;Mean of the nearest neighbors: This approach fills in the missing points with the average value for the k nearest neighbors. To select the optimum K, which corresponds to the number of neighbors considered, we run the algorithm several times with different values of K, choosing the value that reduces the number of errors committed.

The best technique was chosen by measuring the Root Mean Square Error (RMSE), one of the most used criteria to evaluate performance of interpolation methods found in literature [21]. The RMSE determines the standard deviation of the residuals, which corresponds to the difference between the original and the predicted values, where a smaller RMSE implies greater accuracy. Therefore, according to this criterion, the best method is PCHIP, with an RMSE of 1.14.

#### 2.5.2. Modeling

After applying the best filling method found as described on the last section, we were ready to try to approximate the data obtained from Fitbit to the data coming from Polar H10.

Supervised learning problems can be further grouped into regression and classification problems. Both problems have, as a goal, the construction of a succinct model that can predict the value of the dependent variable from the attribute variables. The difference between the two tasks is the fact that the dependent attribute is numerical for regression and categorical for classification. Our predictor variable (HR) is neither continuous nor categorical, but an integer number, which is a discrete value. To guarantee that the best approach was chosen different regression and classification methods were compared:Multinomial logistic regression: A logistic regression that admits for more than two categories for the dependent variable. Here, the discrete numerical values of our dependent variable were considered as categories;K nearest neighbors: a method that assigns a predicted value to a new observation based on the mean of its nearest neighbors;Decision tree: a tree structure in which the data are continuously split according to a certain parameter, where the internal nodes represent the features of the dataset, the branches represent the decision rules, and the leaf nodes represent the outcome;Random forest: a method for classification and regression that works by constructing a large amount of decision trees;AdaBoost: an ensemble machine learning algorithm that combines multiple predictions from many weak learners into a single stronger prediction;Linear regression: an algorithm that assumes a linear relationship between the input variables and the output, attempting to model the relationship between the two variables by fitting a linear equation to the observed data;Neural Networks: Neural Networks are a supervised learning algorithms that seek to approximate a function which is represented by the input data. The type of artificial neural network to handle with sequential data or time series data are a recurrent neural network (RNN). Here we compared traditional single-layer RNN with a Long Short-Term Memory network (LSTM), a special kind of RNN which is capable of learning long-term dependencies.

The times series data were split into train and test, where 75% of the data were used for training the models, with the remaining 25% as a test set for evaluating the final performance of the models. The division was stratified by subject so that we can later calculate the HRV metrics for each individual.

To obtain a more accurate representation of how well our model can perform on data that have not been used before, we used k-fold cross-validation for the hyperparameter tuning. With this method, we split the training data into training and validation multiple times, each time for a different subset of the training data, assigning a score to the model after each iteration and averaging all scores to obtain a better representation of how the model performs. K was set to 5. The data splitting process is detailed in Figure 10.

The hyperparameters of machine learning algorithms are coefficients that allow the model to adapt its behavior to a specific dataset, optimizing the results. Except for linear regression, which does not offer much room for tuning, all these algorithms have a different set of hyperparameters. We used the cross-validation technique to look for the optimal hyperparameter values for each model as depicted for most models in Figure 11, where the best value for each hyperparameter is the one with the highest negated Mean Squared Error (MSE). All scikit-learn scorer objects follow the convention that higher return values are better; therefore, metrics that measure the distance between the prediction and the original data, such as MSE are available as their corresponding negated value of the metric.

The tuning for the Neural Networks is slightly different. Here, cross validation was employed to evaluate each model using Grid Search, a model hyperparameter optimization technique that constructs and evaluates the model for different combinations of hyperparameters. The best accuracy was achieved for a batch size of 64 and 15 epochs. The best results were obtained with the Adam optimizer, a learning rate of 0.1 and a momentum of 0.5. Further details about the Neural Networks’ models as the number of layers, neurons and activation functions are depicted in Figure 12. The LSTM model consists of a LSTM layer and two dense layers which have a sigmoid as the activation function. Dropouts were added between layers and on the LSTM layer to avoid overfitting. On its turn, the RNN model corresponds to a Single Layer Recurrent Neural Network.

Selecting an optimum number of epochs is an important step, since too many epochs can lead to overfitting, whereas too few may result in an underfit model. Given that the goal of training is to minimize the loss, Figure 13 helped us to confirm the results provided by Grid Search for selecting the number of epochs and decide when to stop training, according to when the chosen metric, which in this case is the MSE stopped improving.

After finding optimal hyperparameters and evaluating the model’s average performance with cross validation, we can evaluate the model’s final performance. In order to perform this, we trained the models on the entire 75% of the data that we used for all of our evaluations, using the optimal hyperparameters found for each model and then compare how our models perform on the test set.

For evaluating the model’s final performance, the RMSE was calculated as presented on the Table 1.

Although LSTM and RNN have a similar performance, LSTM has the lowest RMSE, which implies that it can fit the dataset is the best out of all models.

Once the time series is completely filled, the IBI can be calculated from the HR data using the equation [22]:HR = 6000/IBI.

Then the IBI data must be adjusted before any HRV analysis method can be applied, since any artifact in the time series can significantly interfere in the analysis. This adjustment consists of removing inconsistencies such as outliers and ectopic beats (irregular heartbeats). The HRV-analysis library provides different methods to remove these inconsistencies, such as:Malik Rule: IBIs differing by more than 20% from the one preceding it are removed;Karlsson Rule: IBIs diverging by more than 20% of the mean of previous and next IBI are removed;Kamath Rule: this method considers a heartbeat abnormal whenever the IBI increased by more than 32.5% or decreased by more than 24.5% when compared with the previous IBI;Acar Rule: IBIs differing by more than the 20% of the mean of last nine IBIs are removed.

In order to choose the best method, we applied all the above-mentioned approaches to the IBI series obtained from Polar H10 and compared the results with the achieved by Kubios HRV (kubios.com, (accessed on 23 January 2022)) correction methods [23]. Kubios is a heart rate variability analysis software which is already consolidated in the market and in the academy. Since we cannot retrieve the amended IBI series directly from Kubios, we chose the best method by selecting the one which had the HRV indices closer to the results provided by Kubios. At this stage, the identified artefacts were replaced with interpolated values using a cubic spline interpolation, aiming here to approximate the results to those obtained by Kubios, which adopts the same interpolation method [24]. To evaluate the performance of each method, we computed the R Square (R2), which corresponds to the square of the Correlation Coefficient (R) and explains how much variability in dependent variable can be explained by each model, as shown in Figure 14.

R Squared value goes from 0 to 1, where a higher value indicates a better fit between prediction and actual value. Therefore, the method adopted for this study was the Kamath Rule, proposed by [25], once it provided the most similar results to Kubios, showing slightly better results than the other methods.

After all corrections were made, we obtained the mean and its respective standard deviation for each IBI, in a provided time interval, which through mathematical techniques unfold in some statistical indices that constitute the HRV analysis in the time-domain such as the previously defined SDNN, RMSSD and pNN50.

Additionally, the library was extended to include indices such as SDNNi and SDANN, which are calculated considering 5 min segments. Both are important and commonly used measures when taking into account 24-h recordings, which serve as the benchmark for clinical HRV assessment, achieving a greater predictive power than short-term evaluations.

The long-term recording (24 h) supports the healthcare professional in assessing reactions of their patients’ autonomic nervous system during normal daily activities and in response to therapeutic interventions, by comparing the patient’s daily readings to determine if any significant changes have taken place [26].

#### 2.5.3. Model’s Execution

The model execution experiment is illustrated in Figure 15. When using the Fitbit Inspire HR, the elderly have their data stored in the Fitbit server. Then, in (1) the SMH App obtains the user data and provides engagement for the patient or caregiver. After that, in (2) the SMH platform runs smart algorithms that calculate HRV parameters, such as SDNN, RMSSD, and pNN50, based on the patient data. The SMH HRV algorithms reuse parts of the HRV-analysis library (https://pypi.org/project/hrv-analysis, (accessed on 23 January 2022)) that is distributed under the GPLv3 license. In (3) we have the SMH web dashboard consuming data from the Sênior Mobile Health server. Python scripts were written to extract in (4) information about HRV parameters and compared with the parameters calculated by the Kubios HRV tool in (5). Results are discussed further.

## 3. Results

In the previous sections, we validated each step of the analysis workflow individually. For validating the process as a whole, we compared the HRV time-domain indexes obtained by the SMH platform using the data collected from Fitbit Inspire HR with the ones given by Kubios, for the Polar H10 data. Figure 16, Figure 17 and Figure 18 show the distribution of each one of the chosen HRV indexes. The comparison of the boxplots for each index shows that the data are distributed in a similar way. The boxes overlap and the median lines lie within the overlap between boxes indicating that there is no significant difference between the two groups.

As we wish to evaluate paired observations on a single group of patients, we performed a paired Student’s *t*-test (Table 2) to determine whether the paired observations are significantly different from one another.

Using a significance level of 0.05, with a confidence level of 95%, and 9 degrees of freedom (n − 1), the critical value of t obtained from the t distribution table is 1.833. For a confidence level of 99%, recommended for health applications, the critical value is 2.821.

The *t* statistic for the SDNN was 1.772, which is smaller than the critical value for both 95% and 99% confidence levels. The *p*-value was 0.110, which is greater than the significance level, so we accept the null hypothesis that there is no difference in the means of the SDNN metric calculated by SMH. For RMSSD, the obtained *t* statistic was 1.150, which is also smaller than the critical values. The *p*-value was 0.280, that is greater than the significance level, so we accept the null hypothesis that there is no difference between the means. Lastly, for pNN50, the t-statistic was 0.025, which is also smaller than the critical values and the *p*-value was 0.980, that is greater than the significance level, therefore, we accept the null hypothesis.

## 4. Discussion

The results indicate that the time domain HRV indices estimated for Fitbit data by SMH were not statistically significantly different (*p* > 0.05 and *p* > 0.01) from the ones obtained for Polar H10 data by Kubios, for individuals during rest; making it a reasonable alternative for HRV estimation when ECG measurement is impractical.

This analysis reinforces the claims of [14] that PPG is a viable alternative for heartbeat interval measurements for HRV analysis in healthy subjects at rest with significant correlations above 95% for time-domain features, further providing coverage for a larger population by including individuals with comorbidities.

The findings of this study open opportunities to remotely measure and track heart rate variability through telehealth technologies, strengthening strategies for early identification of adverse health conditions and conducting appropriate interventions, that seek to promote the health of the elder population. Several solutions have been implemented to assist in the process of monitoring the health conditions of individuals that corroborates with the practical findings of this work.

Studies using wearable devices, with PPG technology, have been developed as a way to reduce the difficulties encountered in measurements performed using instruments such as ECG. In the study developed in [27], the use of a wearable device was proposed as a method of analysis of HRV, presenting, through the insertion of this technology, the possibility of daily monitoring and outside the outpatient and hospital context of HRV in the elderly. In addition to proposing the use of PPG for HRV analysis, the study associated HRV indicators with physical function in 77 elderly people who were part of the sample, presenting a correlation between these data and introducing the hypotheses of HRV as biomarker of the physical health of the elderly. This attests the importance of more research on the use of the PPG technology in heart rate variability, one of the contributions of the SMH.

In the study developed in [28], with 8 million individuals using PPG data to measure HRV through the Fitbit device, it was possible to measure several metrics of autonomic cardiac health, through HRV indicators, showing that it is a potential tool that can be used in continuous monitoring of individuals. In addition, the study correlated HRV indicators with age and physical activity levels by capturing data through the movement sensors present in the device. In the study, it was possible to observe changes in HRV with advancing age, and with the level of physical activity or sedentary behavior presented by individuals.

In the study conducted by [29], with 49 individuals between 40 and 75 years of age, an evaluation was carried out comparing the heart rate data obtained by smartwatches with those obtained by an ECG monitor. This study observed a good correlation be-tween the heart rate obtained by smartwatches and by ECG. It concludes that smartwatches can be useful for monitoring patients and replicated in developing countries.

Further research is needed to assess limitations of this work, such as sample size and the absence of measurements under free-living conditions (walking, practicing exercise, and sleeping), as motion artifacts make the estimation of HR from PPG more challenging. Future studies should take into account the correlation between indicators of frailty such as weight loss, exhaustion, level of physical activity, walking speed, and grip strength to the HRV.

These findings together with ours may open new opportunities to remotely measure and track heart rate variability through telehealth technologies, strengthening strategies for early identification of adverse health conditions and conducting appropriate interventions, seeking to promote the health of this portion of the population.

Our study reinforces the current scenario and contributes to technological development. The COVID-19 pandemic encouraged the need to create strategies to maintain assistance to the individual continuously, but remotely, preserving and monitoring health and disease indicators. In this way, several remote monitoring platforms using the concepts of telehealth were adopted as measures of care for the population [30,31,32].

Therefore, establishing measures and strategies to monitor the health of the elderly using the potential of technology becomes essential in the context of an aging society and, that needs care. In addition, the technological solutions developed during the pandemic and changes in the form of health care for the elderly will undergo continuous and perhaps, permanent changes. In this way, technology will be increasingly inserted in health care practices.

## 5. Conclusions

In this study we reported the challenges and strategies in the process of building intelligent models to fill in missing data from heart rate time series and the methods of implementing heart rate variability classifiers, acquired using a commercial wearable device, namely Fitbit. The results were compared with complete time series as the gold standard, obtained from Polar H10 chest strap. Following a workflow, we explored several different conventional time series imputation methods in the data pre-processing phase, and different regression and classification methods in the modeling phase to approximate the data obtained from Fitbit to the data coming from Polar H10. In the model’s execution phase, we also presented the feasibility of using this type of technology as a method of long-term monitoring of HRV, focusing on the most important HRV metrics, being a useful for remote monitoring of older persons, enabling the early identification of possible adverse health outcomes and risks that may arise during the human aging process that comprise changes in HRV indicators. Finally, we conclude that the application of the methods and techniques used and reported in this study are useful for the creation and validation of HRV indicators in time series obtained by means of wearable devices based on photoplethysmography sensors.

## Figures and Tables

**Figure 1 sensors-22-02001-f001:**
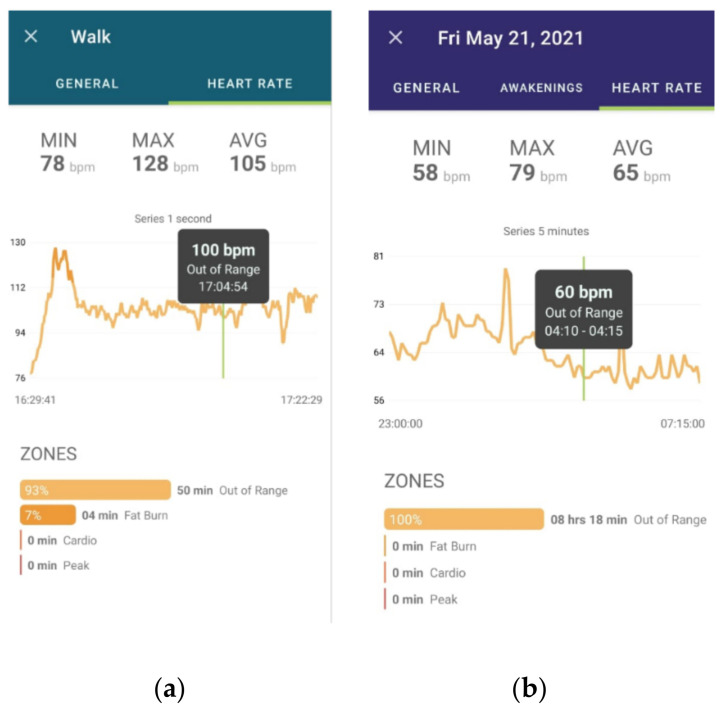
Overview of the SMH mobile app. (**a**) is a walk screen and (**b**) is a sleep screen.

**Figure 2 sensors-22-02001-f002:**
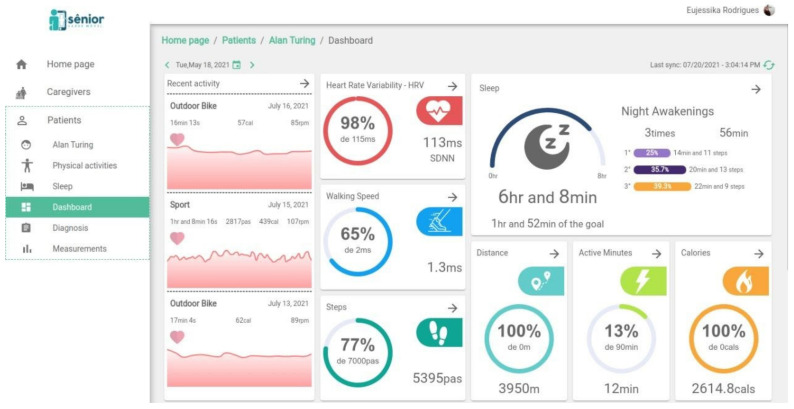
Overview of the SMH web dashboard.

**Figure 3 sensors-22-02001-f003:**
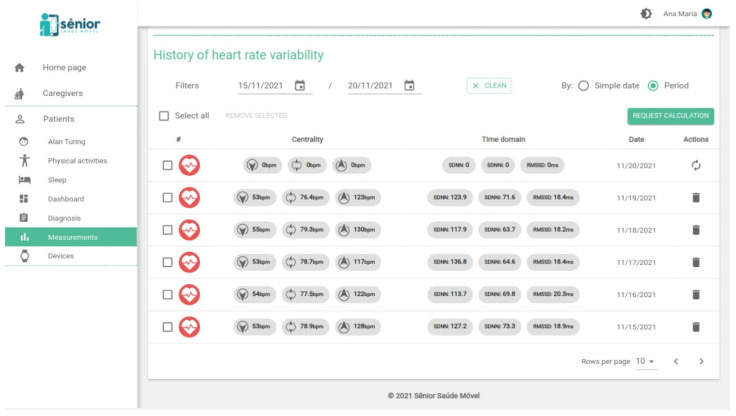
History of heart rate variability screen.

**Figure 4 sensors-22-02001-f004:**
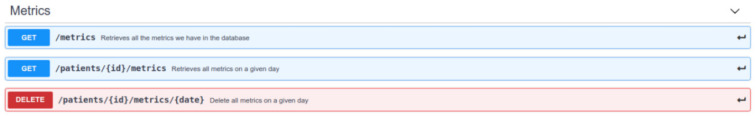
Excerpt of the swagger documentation of the HRV metrics.

**Figure 5 sensors-22-02001-f005:**
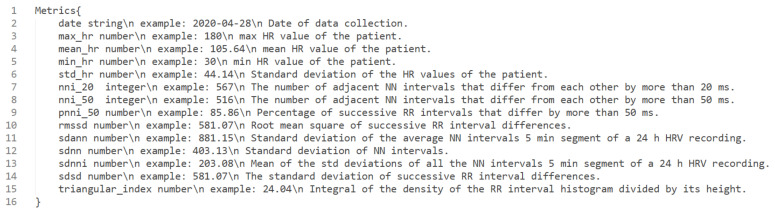
Metrics data model for HRV.

**Figure 6 sensors-22-02001-f006:**
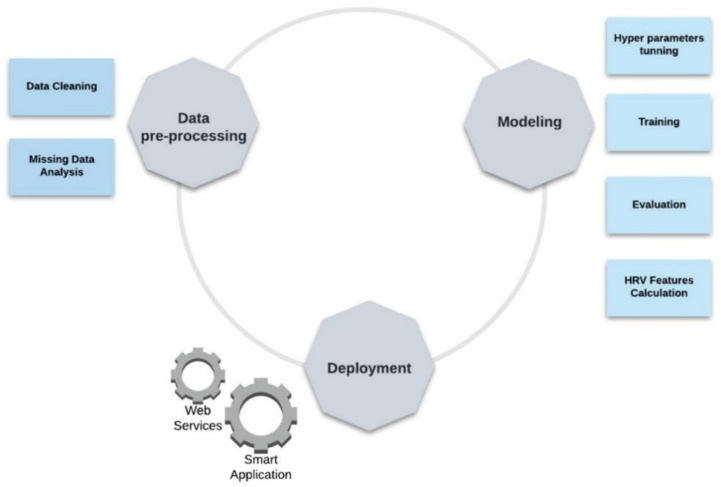
Model overview.

**Figure 7 sensors-22-02001-f007:**
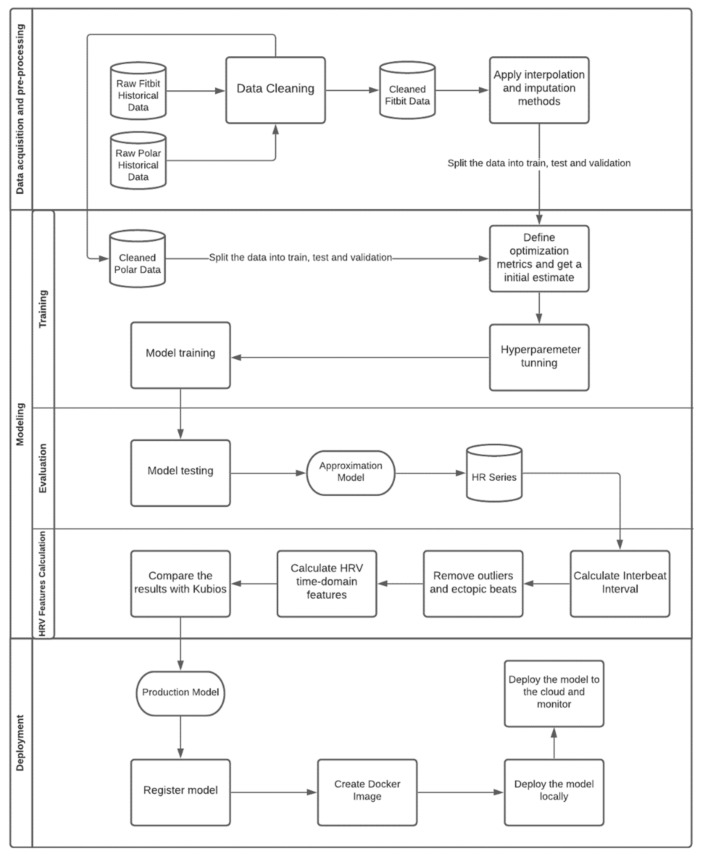
Development process workflow.

**Figure 8 sensors-22-02001-f008:**
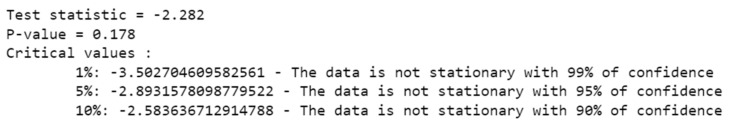
ADF Test.

**Figure 9 sensors-22-02001-f009:**
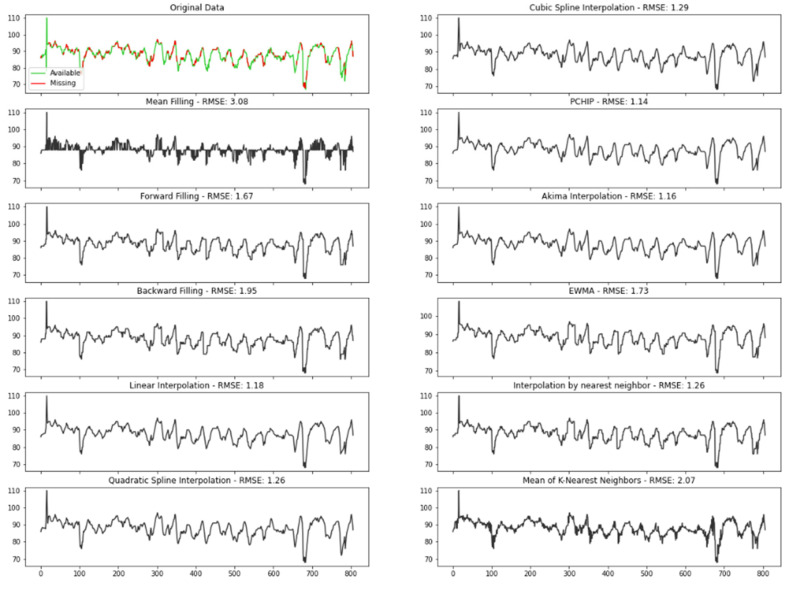
Comparison of the interpolation methods.

**Figure 10 sensors-22-02001-f010:**
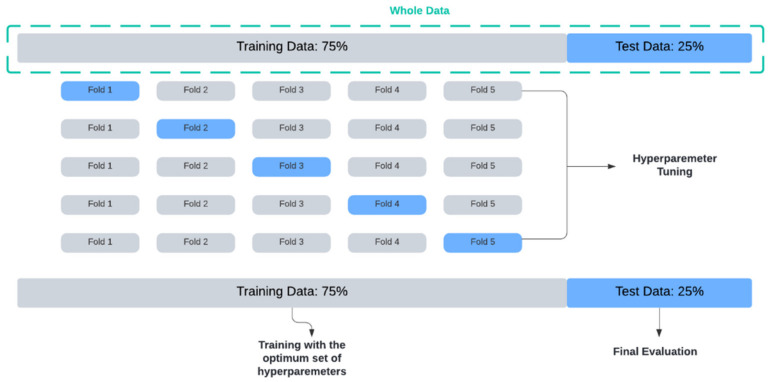
Data splitting.

**Figure 11 sensors-22-02001-f011:**
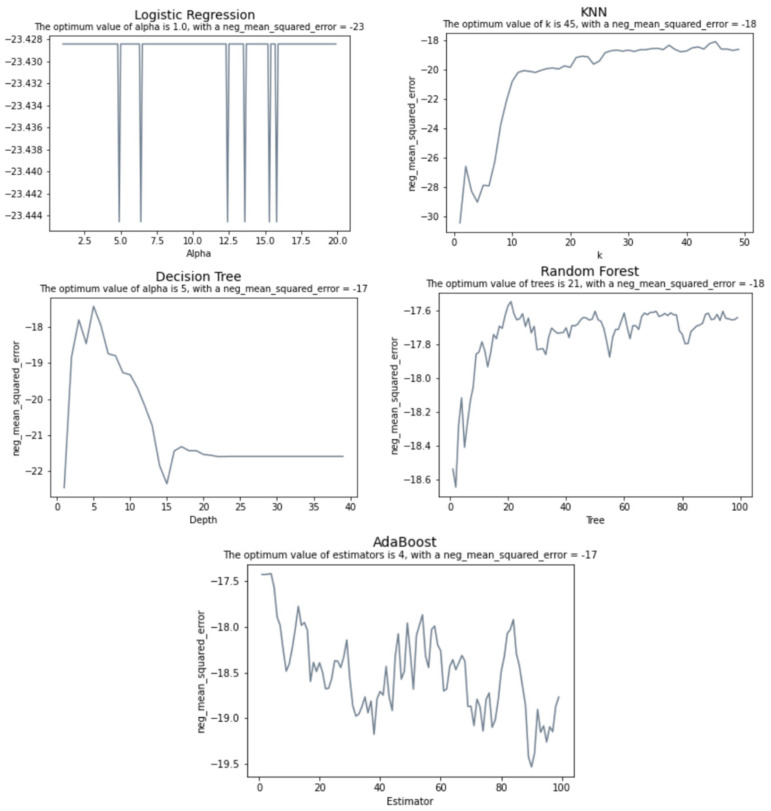
Hyperparameter tuning.

**Figure 12 sensors-22-02001-f012:**
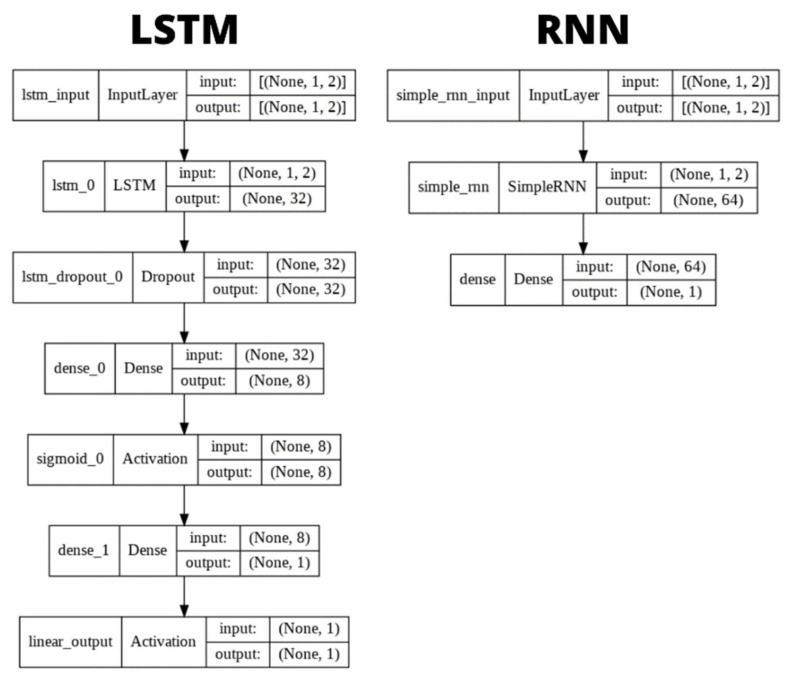
Neural Networks’ models.

**Figure 13 sensors-22-02001-f013:**
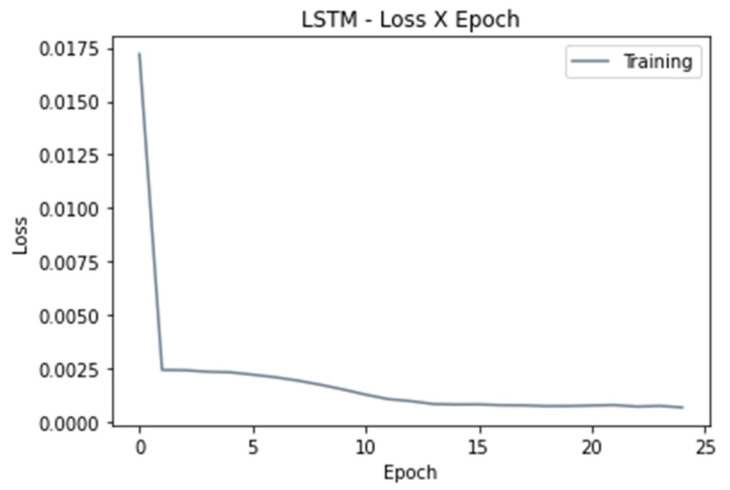
Loss versus number of epoch.

**Figure 14 sensors-22-02001-f014:**
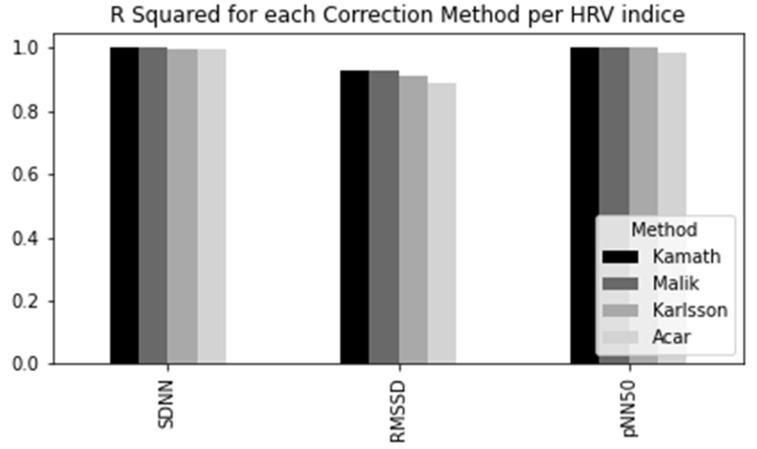
Comparison of the correction methods.

**Figure 15 sensors-22-02001-f015:**
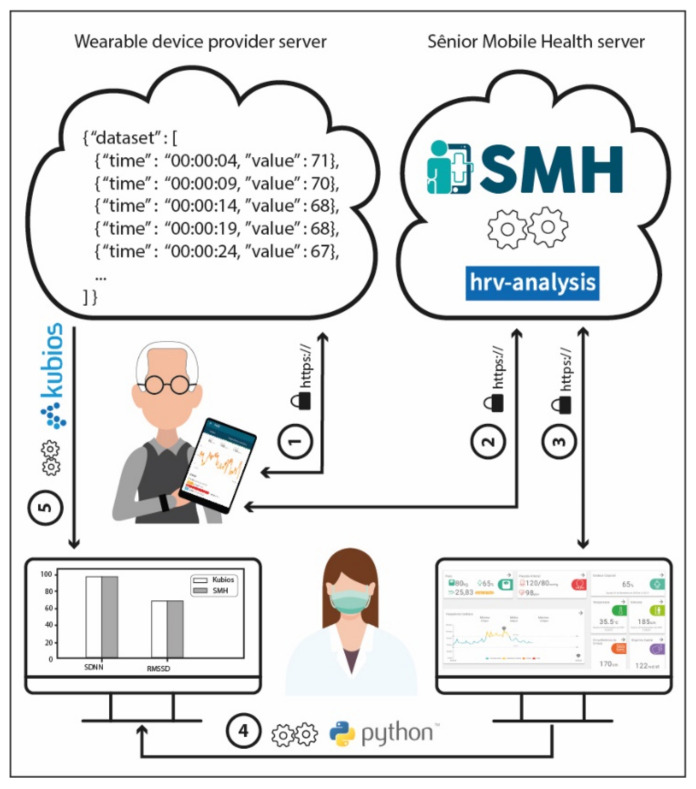
Information flow of the experiment to validate HRV parameters in the SMH platform.

**Figure 16 sensors-22-02001-f016:**
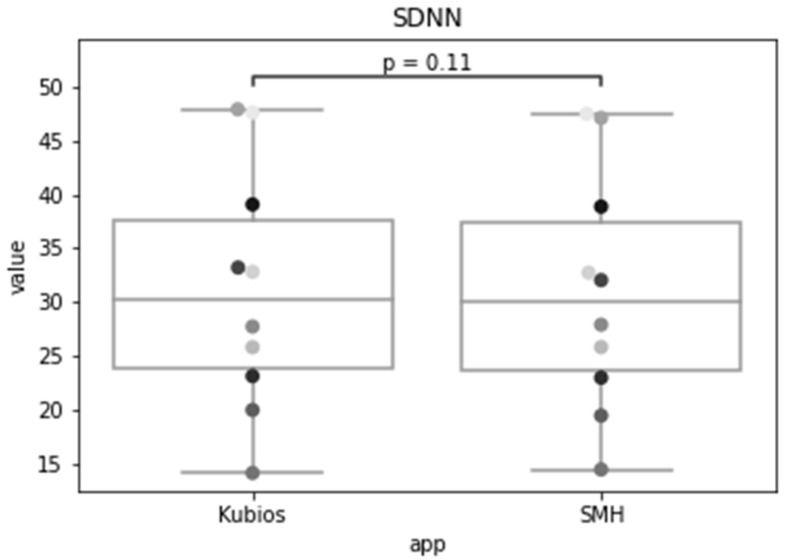
Standard deviation of the of all normal RR intervals (SDNN) of the same group of patients for Kubios and SMH.

**Figure 17 sensors-22-02001-f017:**
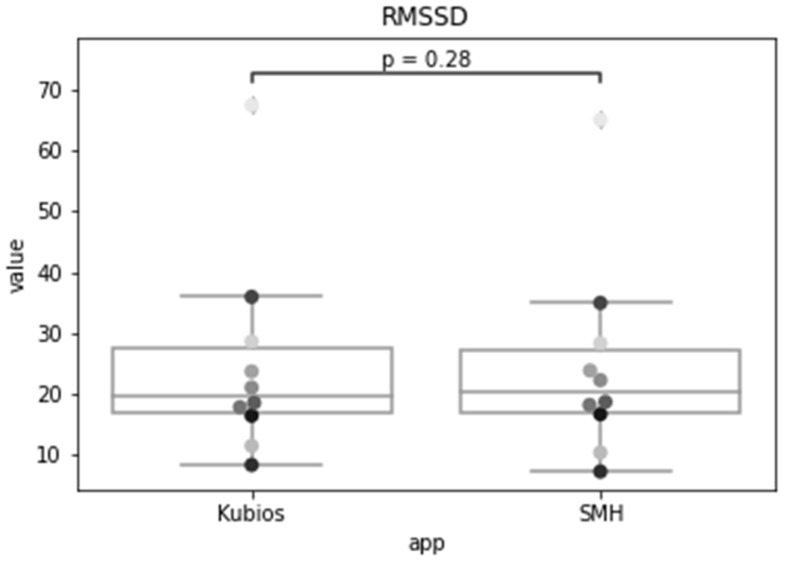
The root mean square of successive differences between normal heartbeats (RMSSD) for Kubios and SMH.

**Figure 18 sensors-22-02001-f018:**
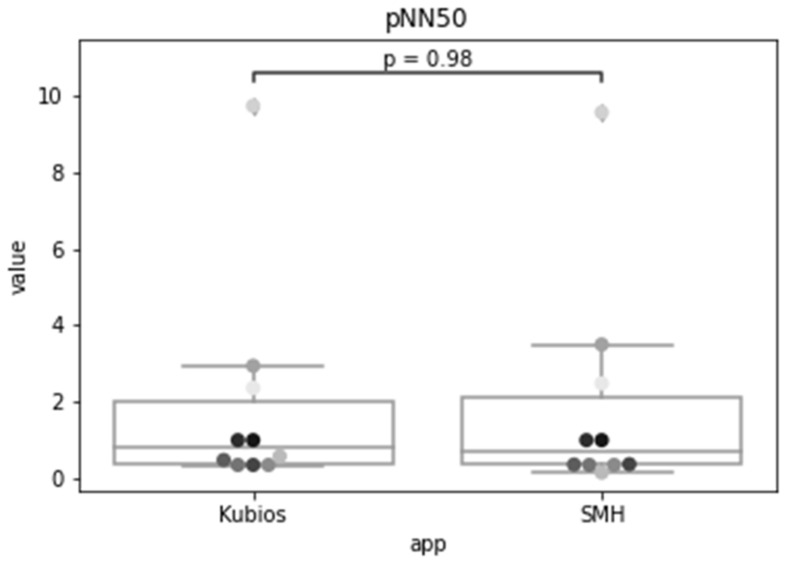
Results for the percentage of successive RR intervals that differ by more than 50 ms (pNN50) using Kubios and SMH.

**Table 1 sensors-22-02001-t001:** Approximation model’s performance.

Algorithm	RMSE
Logistic Regression	4.95
KNN	4.01
Decision Tree	4.07
Random Forest	3.83
AdaBoost	3.91
Linear Regression	2.80
RNN	2.30
LSTM	2.23

**Table 2 sensors-22-02001-t002:** Paired Student’s *t*-test to check if there is any difference in the observations obtained for Fitbit data using SMH in comparison with Polar H10 data on Kubios.

**Null Hypothesis:** There is no difference in the means of the metric calculated by SMH
**Alternate Hypothesis:** There is a difference in the means of the metric calculated by SMH
Metric	t-value	*p*-value	
SDNN	1.772	0.110	
RMSSD	1.150	0.280	
pNN50	0.025	0.980

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
