# Peer review of "HRV Monitoring Using Commercial Wearable Devices as a Health Indicator for Older Persons during the Pandemic"

_sensors, 2022, doi:10.3390/s22052001_

Round 1

Reviewer 1 Report

The article describes a solution for monitoring heart rate variability (HRV) of older people. The proposed solution is based on a “Fitbit Inspire HR” device placed on the wrist, and is a part of the Senior Mobile Health platform. The availability of the API for the device was one of the main factors in its selection. During the research, the “Polar H10” heart monitor and Kubios software were used as a reference. The signal from the device goes through the following stages of the analysis: filling missing values (10 methods were tested, the best was PCHIP), approximation of the obtained data to the data from the Polar H10 (10 methods were tested, LSTM neural network was the best), inter-beat interval (IBI) data adjustment (4 methods were tested, Kamath Rule was the best). Three metrics describing HRV were considered: SDNN, RMSSD, pNN50.

The paper presents good quality, detailed comments are given below:

- the terms “pipeline”, “consume data” seem not very good, also English is worth some improvement

- in line 94 probably should be “Section 5”, in lines 32, 603 probably should be “photoplethysmography”

- in line 262: should be “SMH” instead of “SSM”?

- in line 268: is the reference to “Figure 8” correct?

- in Table 1: is it necessary to present values with such high precision?

- there is no “Section 2.5.3”

- in lines 21, 93 it is written about various “databases”, but in the rest of the work there is no information about them

It should be emphasized positively:

- the authors rightly note the disadvantages in the form of a small number of people investigated in the research and the absence of measurements under free-living conditions

- a large number of figures in the paper

- statistical significance analysis in Table 2

The above comments do not affect the generally high quality of the paper. In my opinion, after these corrections, the article may be published in the Sensors Journal.

Author Response

Thank you in advance for your interest in our paper.

Best regards,

Eujessika Rodrigues.

Reviewer 2 Report

The paper aims to present results from the use of a e-health application to monitor elderly patients using data from a commercial wearable device.  The authors focus on the use of PPG sensor readings to estimate several heart rate variability (HRV) metrics. The estimations are then compared to the ones obtained with the combination of a more complex/precise sensor+software used as gold standard.

The idea of the paper seems sound and interesting. The general description of the whole system is appreciated, experiments are reasonable and results well analyzed.

However, in my opinion the paper can be improved in certain aspects in order to make it more useful for other researchers.

  • Several acronyms are used early in the paper and only defined much later (or never), this hurts the readability by those not already familiar with the terms. For example: PCHIP, SDNN, RMSSD, pNN50,IBI among others.
  • Hyperparameter tuning for the tested ML models is given a cursory mention, with only an example given for KNN (with a questionable conclusion). This step must be detailed to a higher degree, since it is important to determine the solution space explored by the authors and it also helps with repeatability. In particular, on lines 424-425 the authors seem to imply that the selected model is a LSTM network, however there is no clue as to the specific configuration used for it (training time window, number of layers, h vector size, etc) (or for any other model besides KNN).
  • The authors say to have divided total data into training (75%) and testing (25%) plus k-fold cross-validation (k=5). But there is no mention as to how the partition is made, e.g. randomly across the whole dataset? stratified by subject? or, given that the data is used as a time series, do you use the first 75% samples of each recording as train and the rest as test? (in which case, how do you apply k-fold cv?) 
  • On lines 292-297 the authors state that your method should mitigate errors due to the inaccuracy of PPG sensors, especially during periods of exercise. However, the authors fail to provide any evidence as to why this is so (or how much success they actually had with respect to this particular statement) given that all their data is taken in rest conditions. Authors mention something about this in future work, so I think the affirmation about their method currently compensating for activity error must be eliminated until new evidence is provided. Currently their results are only valid for resting conditions.
  • The t-test presented in section 3 uses a 95% confidence level, I suggest testing also at 99% level, given that 95% (although commonly used) is currently not considered stringent enough in many health applications.

Author Response

(The authors gave the same response as above.)
